# EgoMap: Projective mapping and structured egocentric memory for Deep RL

## Abstract

Tasks involving localization, memorization and planning in partially observable 3D environments are an ongoing challenge in Deep Reinforcement Learning. We present EgoMap, a spatially structured neural memory architecture. EgoMap augments a deep reinforcement learning agent's performance in 3D environments on challenging tasks with multi-step objectives. The EgoMap architecture incorporates several inductive biases including a differentiable inverse projection of CNN feature vectors onto a top-down spatially structured map. The map is updated with ego-motion measurements through a differentiable affine transform. We show this architecture outperforms both standard recurrent agents and state of the art agents with structured memory. We demonstrate that incorporating these inductive biases into an agent's architecture allows for stable training with reward alone, circumventing the expense of acquiring and labelling expert trajectories. A detailed ablation study demonstrates the impact of key aspects of the architecture and through extensive qualitative analysis, we show how the agent exploits its structured internal memory to achieve higher performance.

## 1 Introduction

A critical part of intelligence is navigation, memory and planning. An animal that is able to store and recall pertinent information about their environment is likely to exceed the performance of an animal whose behavior is purely reactive. Many control problems in partially observed 3D environments involve long term dependencies and planning. Solving these problems requires agents to learn several key capacities: *spatial reasoning* — to explore the environment in an efficient manner and to learn spatio-temporal regularities and affordances. The agent needs to autonomously navigate, discover relevant objects, store their positions for later use, their possible interactions and the eventual relationships between the objects and the task at hand. Semantic mapping is a key feature in these tasks. A second feature is *discovering semantics from interactions* — while solutions exist for semantic mapping and semantic SLAM Civera et al. (2011); Tateno et al. (2017), a more interesting problem arises when the semantics of objects and their affordances are not supervised, but defined through the task and thus learned from reward.

A typical approach for these types of problems are agents based on deep neural networks including recurrent hidden states, which encode the relevant information of the history of observations Mirowski et al. (2017); Jaderberg et al. (2017). If the task requires navigation, the hidden state will naturally be required to store spatially structured information. It has been recently reported that spatial structure as inductive bias can improve the performance on these tasks. In Parisotto & Salakhutdinov (2018), for instance, different cells in a neural map correspond to different positions of the agent.

In our work, we go beyond structuring the agent's memory with respect to the agent's position. We use projective geometry as an inductive bias to neural networks, allowing the agent to structure its memory with respect to the locations of objects it perceives, as illustrated in Figure 1b. The model performs an inverse projection of CNN feature vectors in order to map and store observations in an egocentric (bird's eye view) spatially structured memory. The EgoMap is complementary to the hidden state vector of the agent and is read with a combination of a global convolutional read operation and an attention model allowing the agent to query the presence of specific content. We show that incorporating projective spatial memory enables the agent to learn policies that exceed the performance of a standard recurrent agent. Two different objects visible in the same input image

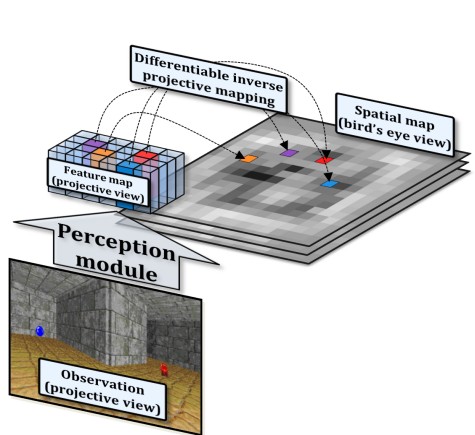 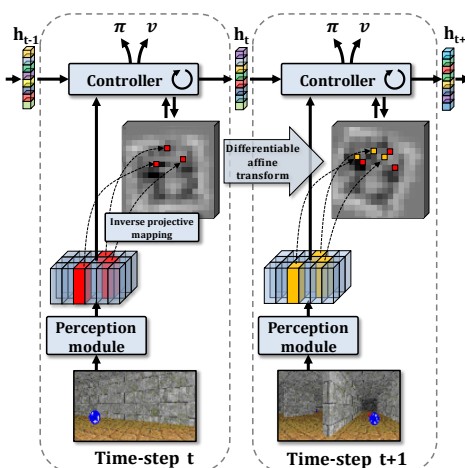

(a) Visual features are mapped to the top-down egocentric map. Observations from a first-person viewpoint are passed through a perception module, extracted features are projected with the inverse camera matrix and depth buffer to their 3D coordinates. The operations are implemented in a differentiable manner, so error derivatives can be back-propagated to update the weights of the perception module. Different objects in the same image are mapped to different locations in the map.

(b) An agent exploring in a 3D environment perceives from a projective egocentric viewpoint. Our model learns to unproject learned task-oriented semantic embeddings of observations and to map the positions of relevant objects in a spatially structured (bird's eye view) neural memory, where different objects in the same input image are stored in different map locations. Using the ego-motion of the agent, the map is updated by differentiable affine resampling at each step in the environment.

Figure 1: Overview of the perception and mapping module (a) and EgoMap agent architecture (b).

could be at very different places in the environment. In contrast to Parisotto & Salakhutdinov (2018), our model will map these observations to their respective locations, and not to cells corresponding to the agent's position, as shown in Figure 1a.

The model bears a certain structural resemblance with Bayesian occupancy grids (BOG), which have been used in mobile robotics for many years Moravec (1988); Rummelhard et al. (2015). As in BOGs, we perform inverse projections of observations and dynamically resample the map to take into account ego-motion. However, in contrast to BOGs, our model does not require a handcrafted observation model and it learns semantics directly from interactions with the environment through reward. It is fully differentiable and trained end-to-end with backpropagation of derivatives calculated with policy gradient methods. Our contributions are as follows:

- To our knowledge, we present the first method using a differentiable SLAM-like mapping of visual features into a top-down egocentric feature map using projective geometry while at the same time training this representation using RL from reward.
- Our spatial map can be translated and rotated through a differentiable affine transform and read globally and through self-attention.
- We show that the mapping, spatial memory and self-attention can be learned end-to-end with RL, avoiding the cost of labelling trajectories from domain experts, auxiliary supervision or pre-training specific parts of the architecture.
- We demonstrate the improvement in performance over recurrent and spatial structured baselines without projective geometry.
- We illustrate the reasoning abilities of the agent by visualizing the content of the spatial memory and the self-attention process, tying it to the different objects and affordances related to the task.
- Experiments with noisy actions demonstrate the agent is robust to actions tolerances of up to 10%.

The code will be made publicly available on acceptance.

## 2 RELATED WORK

**Reinforcement learning** — In recent years the field of Deep Reinforcement Learning (RL) has gained attention with successes on board games Silver et al. (2018) and Atari games Mnih et al. (2015). One key component was the application of deep neural networks Lecun et al. (1998) to frames from the environment or game board states. Recent works that have applied Deep RL for the control of an agent in 3D environments such as maze navigation are Mirowski et al. (2017) and Jaderberg et al. (2017) which explored the use of auxiliary tasks such as depth prediction, loop detection and reward prediction to accelerate learning. Meta RL approaches for 3D navigation have been applied by Wang et al. (2016) and Lample & Chaplot (2017) also accelerated the learning process in 3D environments by prediction of tailored game features. There has also been recent work in the use of street-view scenes to train an agent to navigate in city environments Mirowski et al. (2018). In order to infer long term dependencies and store pertinent information about the partially observable environment; network architectures typically incorporate recurrent memory such as Gated Recurrent Units Chung et al. (2015) or Long Short-Term Memory Hochreiter & Schmidhuber (1997).

**Differentiable memory** — Differentiable memory such as Neural Turing Machines Graves et al. (2014) and Differential Neural Computers Graves et al. (2016) have shown promise where long term dependencies and storage are required. Neural Networks augmented with these memory structures have been shown to learn tasks such as copying, repeating and sorting. Some recent works for control in 2D and 3D environments have included structured memory-based architectures and mapping of observations. Neural SLAM Zhang et al. (2017) aims to incorporate a SLAM-like mapping module as part of the network architecture, but uses simulated sensor data rather than RGB observations from the environment, so the agent is unable to extract semantic meaning from its observations. The experimental results focus on 2D environments and the 3D results are limited. Playing Doom with SLAM augmented memory Bhatti et al. (2016) implements a non-differentiable inverse projective mapping with a fixed feature extractor based on Faster-RCNN Ren et al. (2015), pre-trained in a supervised manner. A downside of this approach is that the network does not learn to extract features pertinent to the task at hand as it is not trained end-to-end with RL. Fang et al. (2019) replace recurrent memory with a transformer (Vaswani et al. (2017)) attention distribution over previous observation embeddings, to highlight that recurrent architectures can struggle to capture long term dependencies. The downside is the storage of previous observations grows linearly with each step in the environment and the agent cannot chose to discard redundant information.

**Grid cells** — there is evidence that biological agents learn to encode spatial structure. Rats develop grid cells/neurons, which fire at different locations with different frequencies and phases, a discovery that led to the 2014 Nobel prize in medicine O'Keefe & Dostrovsky (1971); Hafting et al. (2005). A similar structure seems to emerge in artificial neural networks trained to localize themselves in a maze, discovered independently in 2018 by two different research groups Cueva & Wei (2018); Banino et al. (2018).

**Projective geometry and spatial memory** — Our work encodes spatial structure directly into the agent as additional inductive bias. We argue that projective geometry is a strong law imposed on any vision system working from egocentric observations, justifying a fully differentiable model of perception. To our knowledge, we present the first method which uses projective geometry as inductive bias while at the same time learning spatial semantic features with RL from reward.

The past decade has seen an influx of affordable depth sensors. This has led to a many works in the domain reconstruction of 3D environments, which can be incorporated into robotic systems. Seminal works in this field include Izadi et al. (2011) who performed 3D reconstruction scenes using a moving Kinect sensor and Henry et al. (2014) who created dense 3D maps using RGB-D cameras.

Neural Map Parisotto & Salakhutdinov (2018) implements a structured 2D differentiable memory which was tested in both egocentric and world reference frames, but does not map observations in a SLAM-like manner and instead stores a single feature vector at the agent's current location. The agent's position is also discretized to fixed cells and orientation quantized to four angles (North, South, East, West). A further downside is that the movement of the memory is fixed to discrete translations and the map is not rotated to the agent's current viewpoint.

MapNet Henriques & Vedaldi (2018) includes an inverse mapping of CNN features, is trained in a supervised manner to predict x,y position and rotation from human trajectories, but does not use the

| Related work | Projective Geometry | Spatial map learned from reward | Reinforcement Learning | Imitation learning | Visual input | Multiple goals | Spatially Structured | Task-specific semantic features | Continuous State Space | Continuous Affine transform | End-to-end differentiable | Continuous translations | Continuous rotations | End-to-end training | Intrinsic rewards |
|---|---|---|---|---|---|---|---|---|---|---|---|---|---|---|---|
| Semantic mapping Civera et al. (2011) | ✓ | - | - | - | ✓ | - | ✓ | - | ✓ | ✓ | - | ✓ | ✓ | - | - |
| Playing Doom with SLAM Bhatti et al. (2016) | ✓ | - | ✓ | - | ✓ | - | ✓ | - | ✓ | ✓ | - | ✓ | ✓ | - | - |
| Neural SLAM Zhang et al. (2017) | - | - | ✓ | - | - | - | ✓ | ✓ | ✓ | - | ✓ | - | - | ✓ | - |
| Cog. Map Gupta et al. (2017) | - | - | - | ✓ | ✓ | - | ✓ | ✓ | - | - | ✓ | ✓ | - | ✓ | - |
| Semantic SLAM Tateno et al. (2017) | ✓ | - | - | - | ✓ | - | ✓ | - | ✓ | ✓ | - | ✓ | ✓ | - | - |
| IQA Gordon et al. (2018) | - | - | ✓ | ✓ | ✓ | - | ✓ | - | - | - | - | - | - | - | - |
| MapNet Henriques & Vedaldi (2018) | ✓ | - | - | - | ✓ | - | ✓ | - | ✓ | ✓ | ✓ | ✓ | ✓ | - | - |
| Neural Map Parisotto & Salakhutdinov (2018) | - | ✓ | ✓ | - | ✓ | ✓ | ✓ | ✓ | ✓ | - | ✓ | - | - | ✓ | - |
| MERLIN Wayne et al. (2018) | - | - | ✓ | - | ✓ | ✓ | - | - | ✓ | - | ✓ | ✓ | ✓ | ✓ | - |
| Learning exploration policies Chen et al. (2019) | ✓ | - | ✓ | ✓ | ✓ | - | ✓ | - | ✓ | ✓ | - | ✓ | ✓ | ✓ | ✓ |
| EgoMap | ✓ | ✓ | ✓ | - | ✓ | ✓ | ✓ | ✓ | ✓ | ✓ | ✓ | ✓ | ✓ | ✓ | - |

Table 1: Comparison of key features of related works

map for control in an environment. Visual Question Answering in Interactive Environments Gordon et al. (2018) creates semantic maps from 3D observations for planning and question answering and is applied in a discrete state space.

Unsupervised Predictive Memory in a Goal-Directed Agent Wayne et al. (2018) incorporates a Differential Neural Computer in an RL agent's architecture and was applied to simulated memory-based tasks. The architecture achieves improved performance over a typical LSTM Hochreiter & Schmidhuber (1997) based RL agent, but does not include spatial structure or projective mapping. In addition, visual features and neural memory are learned through the reconstruction of observations and actions, rather than for a specific task.

Cognitive Mapping and Planning for Visual Navigation Gupta et al. (2017) applies a differentiable mapping process on 3D viewpoints in a discrete grid-world, trained with imitation learning which provides supervision on expert trajectories. The downside of discretization is that affine sampling is trivial for rotations of 90-degree increments, and this motion is not representative of the real world. Their tasks are simple point-goal problems of up to 32 time-steps, whereas our work focused on complex multi-step objectives in a continuous state space. Their reliance on imitation learning highlights the challenge of training complex neural architectures with reward alone, particular on tasks with sparse reward such as the ones presented in this paper.

Learning Exploration Policies for Navigation Chen et al. (2019), do not learn a perception module but instead map the depth buffer to a 2D map to provide a map-based exploration reward. Our work learns the features that can be mapped so the agent can query not only occupancy, but task-related semantic content.

Our work greatly exceeds the performance of Neural Map Parisotto & Salakhutdinov (2018), by embedding a differentiable inverse projective transform and a continuous egocentric map into the agent's network architecture. The mapping of the environment is in the agent's reference frame, including translation and rotation with a differentiable affine transform. We demonstrate stable training with reinforcement learning alone, over several challenging tasks and random initializations, and do not require the expense of acquiring expert trajectories. We detail the key similarities and differences with related work in table 1.

## 3   EGOMAP

We consider partially observable Markov decision processes (POMDPs) Kaelbling et al. (1998) in 3D environments and extend recent Deep-RL models, which include a recurrent hidden layer to store pertinent long term information Mirowski et al. (2017); Jaderberg et al. (2017). In particular, RGBD observations $I_t$ at time step $t$ are passed through a perception module extracting features $s_t$, which are used to update the recurrent state:

$$s_t = f_p(I_t; \theta_p) \qquad h_t = f_r(h_{t-1}, s_t; \theta_r) \tag{1}$$

where $f_p$ is a convolutional neural network and $f_r$ is a recurrent neural network in the Gated Recurrent Unit variant Chung et al. (2015). Gates and their equations have been omitted for simplicity. Above and in the rest of this paper, $\theta_*$ are trainable parameters, exact architectures are provided in the appendix. The controller outputs an estimate of the policy (the action distribution) and the value function given its hidden state:

$$\pi_t = f_\pi(h_t; \theta_\pi) \qquad v_t = f_v(h_t; \theta_v) \tag{2}$$

The proposed model is motivated by the regularities which govern 3D physical environments. When an agent perceives an observation of the 3D world, it observes a 2D planar perspective projection of the world based on its current viewpoint. This projection is a well understood physical process, we aim to imbue the agent's architecture with an inductive bias based on inverting the 3D to 2D planar projective process. This inverse mapping operation appears to be second nature to many organisms, with the initial step of depth estimation being well studied in the field of Physiology Frégnac et al. (2004). We believe that providing this mechanism implicitly in the agent's architecture will improve its reasoning capabilities in new environments bypass a large part of the learning process.

The overall concept is that as the agent explores the environment, the perception module $f_p$ produces a 2D feature map $s_t$, in which each feature vector represents a learned semantic representation of a small receptive field from the agent's egocentric observation. While they are integrated into the flat (not spatially structured) recurrent hidden state $h_t$ through function $f_r$ (Equation 1), we propose its integration into a second tensor $M_t$, a top-down egocentric memory, which we call *EgoMap*. The feature vectors are mapped to their egocentric positions using the inverse projection matrix and depth estimates. This requires an agent with a calibrated camera (known intrinsic parameters), which is a soft constraint easily satisfied. The map can then be read by the agent in two ways: a global convolutional read operation and a self-attention operation.

Formally, let the agent's position and angle at time $t$ be $(x_t, y_t)$ and $\phi_t$ respectively, $M_t$ is the current EgoMap. $s_t$ are the feature vectors extracted by the perception module, $D_t$ are the depth buffer values. The change in agent position and orientation in the agent's frame of reference between time-step $t$ and $t-1$ are $(dx_t, dy_t, d\phi_t)$.
There are three key steps to the operation of the EgoMap:

1. Transform the map to the agent's egocentric frame of reference:
$$\hat{M}_t = \text{Affine}(M_{t-1}, dx_t, dy_t, d\phi_t) \tag{3}$$

2. Update the map to include new observations:
$$\tilde{M}_t = \text{InverseProject}(s_t, D_t) \qquad M'_t = \text{Combine}(\hat{M}_t, \tilde{M}_t) \tag{4}$$

3. Perform a global read and attention based read, the outputs of which are fed into the policy and value heads:
$$r_t = \text{Read}(M'_t) \qquad c_t = \text{Context}(M'_t, s_t, r_t) \tag{5}$$

These three operations will be further detailed below in individual subsections. Projective mapping and spatially structured memory should augment the agent's performance where spatial reasoning and long term recollection are required. On simpler tasks the network can still perform as well as the baseline, assuming the extra parameters do not cause further instability in the RL training process.

**Affine transform** — At each time-step we wish to translate and rotate the map into the agent's frame of reference, this is achieved with a differentiable affine transform, popularized by the well known Spatial Transformer Networks Jaderberg et al. (2015). Relying on the simulator to be an oracle and provide the change in position $(dx, dy)$ and orientation $d\phi$, we convert the deltas to the agent's egocentric frame of reference and transform the map with a differentiable affine transform. The effect of noise on the change in position on the agent's performance is analysed in the experimental section.

**Inverse projective mapping** — We take the agent's current observation, extract relevant semantic embeddings and map them from a 2D planar projection to their 3D positions in an egocentric frame of reference. At each time-step, the agent's egocentric observation is encoded by the perception module (a convolutional neural network) to produce feature vectors, this step is a mapping from $R^{4 \times 64 \times 112} \rightarrow R^{16 \times 4 \times 10}$. Given the inverse camera projection matrix and the depth buffer provided by the simulator, we can compute the approximate location of the features in the agent's egocentric frame of reference. As the mapping is a many to one operation, several features can be mapped to the same location. Features that share the same spatial location are averaged element-wise.

The newly mapped features must then be combined with the translated map from the previous time-step. We found that the use of a momentum hyper-parameter, $\alpha$, enabled a smooth blending of new and previously observed features. We use an $\alpha$ value of 0.9 for the tests presented in the paper. We ensured that the blending only occurs where the locations of new projected features and the map from the previous time-step are co-located, this criterion is detailed in Equation 6.

$$M_t'^{(x,y)} = \eta \hat{M}_t^{(x,y)} + (1 - \eta)\tilde{M}_t^{(x,y)}$$

$$\eta = \begin{cases} 1.0, & \text{if } \tilde{M}_t^{(x,y)} = 0 \ \& \ \hat{M}_t^{(x,y)} \neq 0 \\ 0.0, & \text{if } \tilde{M}_t^{(x,y)} \neq 0 \ \& \ \hat{M}_t^{(x,y)} = 0 \\ \alpha, & \text{otherwise} \end{cases} \tag{6}$$

**Sampling from a global map** — A naive approach to the storage and transformation of the egocentric feature map would be to apply an affine transformation to the map at each time-step. A fundamental downside of applying repeated affine transforms is that at each step a bilinear interpolation operation is applied, which causes smearing and degradation of the features in the map. We mitigated this issue by storing the map in a global reference frame and mapping the agent's observations to the global reference frame. For the read operation an offline affine transform is applied. For further details see the appendix B

**Read operations** — We wanted the agent to be able to summarize the whole spatial map and also selectively query for pertinent information. This was achieved by incorporating two types of read operation into the agent's architecture, a *Global Read* operation and a *Self-attention Read*.

The global read operation is a CNN that takes as input the egocentric map and outputs a 32-dimensional feature vector that summarizes the map's contents. The output of the global read is concatenated with the visual CNN output.

To query for relevant features in the map, the agent's controller network can output a query vector $q_t$, the network then compares this vector to each location in the map with a cosine similarity function in order to produce scores, which are the same width and height as the map. The scores are normalized with a softmax operation to produce a soft-attention in the lines of Bahdanau et al. (2014) and used to compute a weighted average of the map, allowing the agent to selectively query and focus on parts of the map. This querying mechanism was used in both the Neural Map Parisotto & Salakhutdinov (2018) and MERLIN Wayne et al. (2018) RL agents. We made the following improvements: *Attention Temperature* and *Query Position*.

$$\sigma(x)_i = \frac{e^{\beta x_i}}{\sum e^{\beta x_j}} \tag{7}$$

*Query Position*: A limitation of self-attention is that the agent can query *what* it has observed but not *where* it had observed it. To improve the spatial reasoning performance of the agent we augmented the neural memory with two fixed additional coordinate planes representing the x,y egocentric coordinate system normalized to $(-1.0, 1.0)$, as introduced for segmentation in Liang et al. (2018). The agent still queries based on the features in the map, but the returned context vector includes two extra scalar quantities which are the weighted averages of the x,y planes. The impacts of these additions are discussed and quantified in the ablation study, in Section 4.

*Attention Temperature*: To provide the agent with the ability to learn to modify the attention distribution, the query includes an additional learnable temperature parameter, $\beta$, which can adjust the softmax distribution detailed in Equation 7. This parameter can vary query by query and is constrained to be one or greater by a Oneplus function. The use of temperature in neural memory architectures was first introduced in Neural Turing Machines Graves et al. (2014).

## 4 EXPERIMENTS

The EgoMap and baseline architectures were evaluated on four challenging 3D scenarios, which require navigation and different levels of spatial memory. The scenarios are taken from Beeching et al. (2019) who extended the 3D ViZDoom environment Kempka et al. (2017) with various scenarios that are partially observable, require memory, spatial understanding, have long horizons and sparse rewards. Whilst more visually realistic simulators are available such as Gibson Xia et al. (2018), Matterport Anderson et al. (2018), Home Brodeur et al. (2018) and Habitat Manolis Savva* et al.

| | Scenario | | | | | | | |
|---|---|---|---|---|---|---|---|---|
| | 4 item | | 6 item | | Find and Return | | Labyrinth | |
| Agent | Train | Test | Train | Test | Train | Test | Train | Test |
| Random | -0.179 | -0.206 | -0.21 | -0.21 | -0.21 | -0.21 | -0.115 | -0.086 |
| Baseline | $2.341 \pm 0.026$ | $2.266 \pm 0.035$ | $2.855 \pm 0.164$ | $2.545 \pm 0.226$ | $0.661 \pm 0.003$ | $0.633 \pm 0.027$ | $0.73 \pm 0.02$ | $0.694 \pm 0.009$ |
| Neural Map | $2.339 \pm 0.038$ | $2.223 \pm 0.040$ | $2.750 \pm 0.062$ | $2.465 \pm 0.034$ | $0.825 \pm 0.070$ | $0.723 \pm 0.026$ | $\mathbf{0.769 \pm 0.042}$ | $0.706 \pm 0.018$ |
| EgoMap | $\mathbf{2.398 \pm 0.014}$ | $\mathbf{2.291 \pm 0.021}$ | $\mathbf{3.214 \pm 0.007}$ | $\mathbf{2.801 \pm 0.048}$ | $\mathbf{0.893 \pm 0.007}$ | $\mathbf{0.848 \pm 0.017}$ | $0.753 \pm 0.002$ | $\mathbf{0.732 \pm 0.016}$ |
| Optimum (Upper Bound) | 2.5 | 2.5 | 3.5 | 3.5 | 1.0 | 1.0 | 1.0 | 1.0 |

Table 2: Results of the baseline and EgoMap architectures trained on four scenarios for 1.2 B environment steps. We show the mean and std. of the final agent performance, evaluated for three independent experiments on a held-out testing set of scenario configurations.

(2019), the tasks available are simple point-goal tasks which do not require long term memory and recollection. We target the following three tasks:

**Labyrinth**: The agent must find the exit in the fastest time possible, the reward is a sparse positive reward for finding the exit. This tests an agent's ability to explore in an efficient manner.

**Ordered k-item**: An agent must find $k$ objects in a fixed order. It tests three aspects of an agent: its ability to explore the environment efficiently, the ability to learn to collect items in a predefined order and its ability to store as part of its hidden state where items were located so they can be retrieved in the correct order. We tested two versions of this scenario with 4-items or 6-items.

**Find and return**: The agent starts next to a green totem, must explore the environment to find a red totem and then return to the starting point. This is our implementation of "Minotaur" scenario from Parisotto & Salakhutdinov (2018). The scenario tests an agent's ability to navigate and retain information over long time periods.

All the tasks require different levels of spatial memory and reasoning. For example, if an agent observes an item out of order it can store the item's location in its spatial memory and navigate back to it later. We observe that scenarios that require more spatial reasoning, long term planning and recollection are where the agent achieves the greatest improvement in performance. In all scenarios there is a small negative reward for each time-step to encourage the agent to complete the task quickly.

**Experimental strategy and generalization to unseen environments** — Many configurations of each scenario were created through procedural generation and partitioned into separated training and testing sets of size 256 and 64 respectively for each scenario type. Although the task in a scenario is fixed, we vary the locations of the walls, item locations, and start and end points; thus we ensure a diverse range of possible scenario configurations. A limited hyper-parameter sweep was undertaken with the baseline architecture to select the hyper-parameters, which were fixed for both the baseline, Neural Map and EgoMap agents. Three independent experiments were conducted per task to evaluate the algorithmic stability of the training process. To avoid information asymmetry, we provide the baseline agent with $dx, dy, sin(d\theta), cos(d\theta)$ concatenated with its visual features.

**Training Details** — The model parameters were optimized with an on-policy, policy gradient algorithm; batched Advantage Actor Critic (A2C) Mnih et al. (2016), we used the popular PyTorch Paszke et al. (2017) implementation of A2C Kostrikov (2018). We sampled trajectories from 16 parallel agents and updated every 128 steps in the environment with discounted returns bootstrapped from value estimates for non-terminal states. The gamma factor was 0.99, the entropy weight was 0.001, the RMSProp Tieleman & Hinton (2012) optimizer was used with a learning rate of 7e-4. The EgoMap agent map size was $16 \times 24 \times 24$ with a grid sampling chosen to cover the environment size with a 20% padding. The agent's policy was updated over 1.2B environment steps, with a frame skip of 4. Training took 36 hours for the baseline and 8 days for the EgoMap, on 4 Xeon E5-2640v3 CPUs, with 32GB of memory and one NVIDIA GK210 GPU.

**Results** — Results from the baseline and EgoMap policies evaluated on the 4 scenarios are shown in table 2, all tasks benefit from the inclusion of inverse projective mapping and spatial memory in the agent's network architecture, with the largest improvement on the *Find and Return* scenario. We postulate that the greater improvement in performance is due to two factors; firstly this scenario always requires spatial memory as the agent must return to its starting point and secondly the objects in this scenario are larger and occupy more space in the map. We also compared to the state of the art in spatially structured neural memory, Neural Map Parisotto & Salakhutdinov (2018). Figure 2 shows agent training curves for the recurrent baseline, Neural Map and EgoMap, on the *Find and Return* test set configurations.

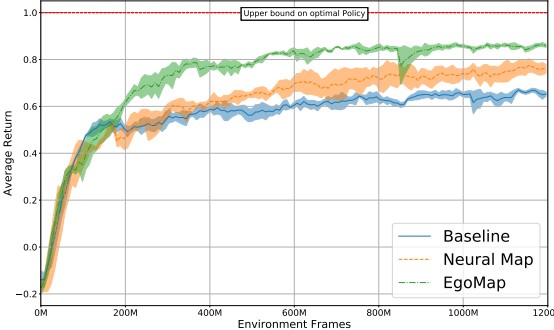

| Ablation | Train | Test |
|---|---|---|
| Baseline | $0.668 \pm 0.028$ | $0.662 \pm 0.036$ |
| No global read | $0.787 \pm 0.007$ | $0.771 \pm 0.029$ |
| No query | $0.838 \pm 0.003$ | $0.811 \pm 0.013$ |
| No query temperature | $0.845 \pm 0.014$ | $0.815 \pm 0.019$ |
| No query position | $0.839 \pm 0.007$ | $0.814 \pm 0.008$ |
| EgoMap | $0.847 \pm 0.011$ | $0.814 \pm 0.017$ |
| EgoMap (L1 query) | $\mathbf{0.851 \pm 0.014}$ | $\mathbf{0.828 \pm 0.011}$ |

Figure 2: Left: Agent performance on unseen test configurations of the *Find and Return* scenario. Right: Ablation study on the *Find and Return* scenario conducted after 800M environment steps.

**Ablation study** — An ablation study was carried out on the improvements made by the EgoMap architecture. We were interested to see the influence of key options such as the global and attention-based reads, the similarity function used when querying the map, the learnable temperature parameter and the incorporation of location-based querying. The Cartesian product of these options is large and it was not feasible to test them all, we therefore decided to selectively switch off key options to understand which aspects contribute to the improvement in performance. The results of the ablation study are shown in Table 2. Both the global and self-attention reads provide large improvements in performance over the baseline recurrent agent. The position-based query provides a small improvement. A comparison of the similarity metric of the attention mechanism highlights the L1-similarity achieved higher performance than cosine. A qualitative analysis of the self-attention mechanism is shown in the next section.

## 5 ANALYSIS

**Noisy Actions** — One common criticism of agents trained in simulation is that the agent can query its environment for information that would not be readily available in the real world. In the case of EgoMap, the agent is trained with ground truth ego-motion measurements. Real-world robots have noisy estimates of ego-motion due to the tolerances of available hardware. We performed an analysis of the EgoMap agent trained in the presence of a noisy oracle, which adds noise to the ego-motion measurement. Noise is drawn from a normal distribution centred at one and is multiplied by the agent's ground-truth motion, the effect of the noise is cumulative but unbiased. Tests were conducted with standard deviations of up to 0.2 which is a tolerance of more than 20% on the agent's ego-motion measurements, results are shown in Figure 3. We observed retain the performance increase over the baseline for up to 10% of noisy actions, the performance degrades to that of the baseline agent.

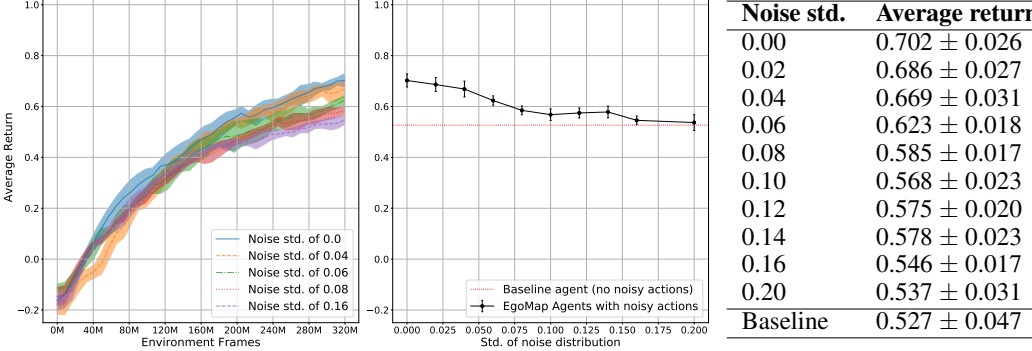

| Noise std. | Average return |
|---|---|
| 0.00 | $0.702 \pm 0.026$ |
| 0.02 | $0.686 \pm 0.027$ |
| 0.04 | $0.669 \pm 0.031$ |
| 0.06 | $0.623 \pm 0.018$ |
| 0.08 | $0.585 \pm 0.017$ |
| 0.10 | $0.568 \pm 0.023$ |
| 0.12 | $0.575 \pm 0.020$ |
| 0.14 | $0.578 \pm 0.023$ |
| 0.16 | $0.546 \pm 0.017$ |
| 0.20 | $0.537 \pm 0.031$ |
| Baseline | $0.527 \pm 0.047$ |

Figure 3: Test set performance of the EgoMap agent during training with noisy ego-motion measurements, conducted for 320 M environment frames. Shown are test set performance during training (left), final performance for a range of noise values (centre) which are tabulated (right). We retain an improvement in performance over the baseline agent for noisy actions of up to 10%.

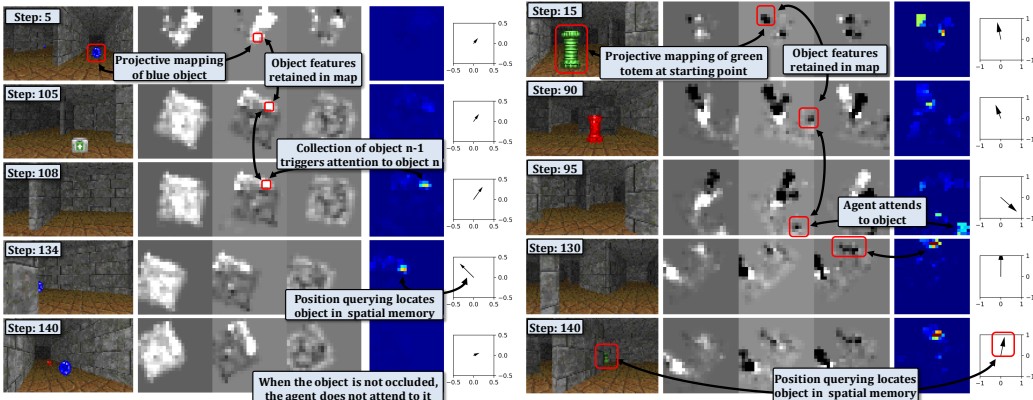

(a) Analysis: *Ordered 6-item* scenario  (b) Analysis: *Find and Return* scenario

Figure 4: Analysis of the EgoMap for key steps (different rows) during an episode from the *Ordered 6-item* and *Find and Return* scenarios. Within each sub-figure: Left column - RGB observations, central column - the three largest PCA components of features mapped in the spatially structured memory, right - attention heat map (result of the query) and x,y query position vector. We observe that the agent maps and stores features from key objects and attends to them when they are pertinent to the current stage of the task. For example, for the left figure on the first row at time-step 5 the blue spherical object, which is ordered 4 of 6, is mapped into the agent's spatial memory. The agent explores the environment collecting the items in order, it collects item 3 of 6 between time-step 105 and 108, shown on rows 2 and 3. As soon as the agent has collected item 3 it queries its internal memory for the presence of item 4, which is shown by the attention distribution on rows 3 and 4. On the last row, time-step 140, the agent observes the item and no longer attempts to query for it, as the item is in the agent's field of view.

**Visualization** — The EgoMap architecture is highly interpretable and provides insights about how the agent reasons in 3D environments. In Figures 4a and 4b we show analysis of the spatially structured memory and how the agent has learned to query and self-attend to recall pertinent information. The Figures show key steps during an episode in the *Ordered 6-item* and *Find and Return* scenarios, including the first three principal components of a dimensionality reduction of the 16-dimensional EgoMap, the attention distribution and the vector returned from position queries. Refer to the caption for further details. The agent is seen to attend to key objects at certain phases of the task, in the *Ordered 6-item* scenario the agent attends the next item in the sequence and in the *Find and Return* scenario the agent attends to the green totem located at the start/return point once it has found the intermediate goal.

## 6 CONCLUSION

We have presented EgoMap, an egocentric spatially structured neural memory that augments an RL agent's performance in 3D navigation, spatial reasoning and control tasks. EgoMap includes a differentiable inverse projective transform that maps learned task-specific semantic embeddings of agent observations to their world positions. We have shown that through the use of global and self-attentive read mechanisms an agent can learn to focus on important features from the environment. We demonstrate that an RL agent can benefit from spatial memory, particularly in 3D scenarios with sparse rewards that require localization and memorization of objects. EgoMap out-performs existing state of the art baselines, including Neural Map, a spatial memory architecture. The increase in performance compared to Neural Map is due to two aspects. 1) The differential projective transform maps *what* the objects are to *where* they are in the map, which allows for direct localization with attention queries and global reads. In comparison, Neural Map writes *what* the agent observes to *where* the agent is on the map, this means that the same object viewed from two different directions will be written to two different locations on the map, which leads to poorer localization of the object. 2) Neural Map splits the map to four 90-degree angles, which alleviates the blurring highlighted in the appendix, our novel solution to this issue stores a single unified map in an allocentric frame of

reference and performs an offline egocentric read, which allows an agent to act in states spaces where the angle is continuous, without the need to quantize the agent's angle to 90-degree increments.

We have shown, with detailed analysis, how the agent has learned to interact with its structured internal memory with self-attention. The ablation study has shown that the agent benefits from both the global and self-attention operations and that these can be augmented with temperature, position querying and other similarity metrics. We have demonstrated that the EgoMap architecture is robust to actions with tolerances of up to 10%. Future work in this area of research would be to apply the mapping and memory architecture in more realistic-looking domains and aim to incorporate both dynamic and static objects into the agent's network architecture and update mechanisms.

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

# A    APPENDIX

# B    AFFINE TRANSFORM

A naive implementation of the repeated affine transforms leads to smearing of features Figure 5 demonstrates the degradation of features on synthetic RGB images with repeated rotations and translations. We should how storing the features in an allocentric frame of reference and performing offline transforms for read operations can greatly mitigate this issue.

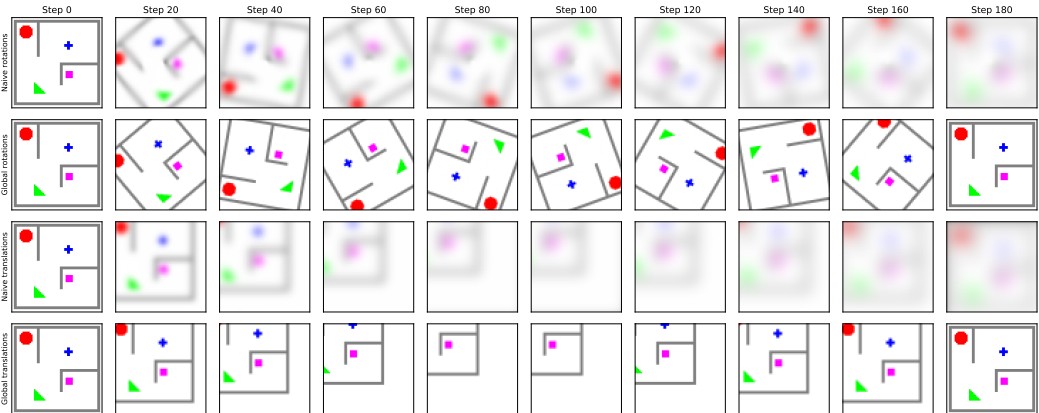

Figure 5: A synthetic 3-channel top down map. Rotations of two degrees per time-step with naive and global reference frame (rows 1 & 2) shown over 180 time-steps. Translations with naive and global reference frame (rows 3 & 4) shown over 180 time-steps. Distinct degradation of the map features are observed by time-step 100 and features are barely visible after 180 steps, leading to poor localization with self-attention queries. In tasks that span up to 500 time-steps, the blurring of features could have greatly restricted the EgoMap agent's performance.

# C    ARCHITECTURES

To encourage reproducibility, we detail the exact architectures of the agents. Figure 6 shows an overview.

**Baseline Model** — is comprised of the following: *Perception Module* $f_p$: A 3 layer CNN with kernel sizes of 8,4,3, strides of 4,2,1, no padding and filter sizes of 16,32,16, respectively and ReLU activation. $f_p$ is a mapping from an RGBD input observation of $R^{4 \times 64 \times 112} \rightarrow R^{16 \times 4 \times 10}$. *Recurrent Module* $f_r$: A FC layer reduces the output of $f_p$ from 640 values to a vector of size 128 and includes a ReLU activation; we then use a GRU layer with 128 hidden units. *Policy and Value Heads* $f_\pi$ & $f_v$: These layers receive as input the output of $f_r$. The policy layer is a FC layer with 5 output units corresponding to the 5 discrete actions available to the agent (through a softmax activation). The value layer is a FC layer with one output unit.

**EgoMap Model** — is comprised of the following: *Perception Module* $f_p$: The same as the baseline architecture, apart from that the mapping operation is applied before the final ReLU activation function. *EgoMap Global Read Module*: A 3 layer CNN with kernel sizes of 3,4,4, strides of 1,2,2 and filter sizes of 16,16,16 respectively, and no padding. Followed by two linear layers of 256 and 32 hidden units, and ReLU activations, apart from the last which was tanh. *Recurrent Module* $f_r$: The output of $f_p$ and the global read module were concatenated to form a vector of size 672 and fed into a FC layer later with 128 output units. The recurrent module was a GRU with 128 units. *Self-Attention Read Head*: The query head is a linear layer with 17 output units, 16 for the calculation of the EgoMap similarity scores and one for the $\beta$ temperature parameter. The query head returns a vector of size 18 which includes two more scalar values for the average position of the query. *Policy*

*and Value Heads* $f_\pi$ *&* $f_v$: Are the same as the baseline but their input is the concatenation of the output of the $f_r$ and the attention head.

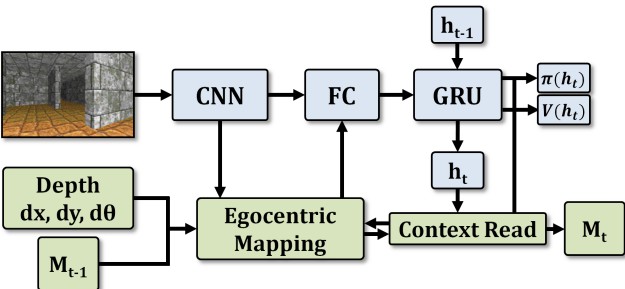

Figure 6: The baseline agent architecture (light blue) augmented with the EgoMap architecture (green), the agent's hidden state is $h_t$ and the spatially structured neural memory $M_t$. We rely on an oracle (the simulator) to provide the depth buffer $D_t$ and the agent deltas $(dx, dy, d\theta)$.

## D READ MECHANISMS

In figure 7 we provide further details of the operation of the global read, context read and xy-querying.

## E ADDITIONAL RESULTS

In figure 8 we provide the curves of agent performance on held out test configurations for three scenarios: 4-item, 6-item and labyrinth.

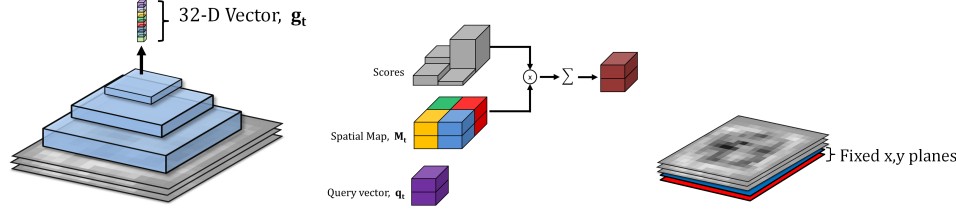

Figure 7: Schematics detailing the operation of the read mechanisms, with global read (left), context read(middle) and xy-querying (right).

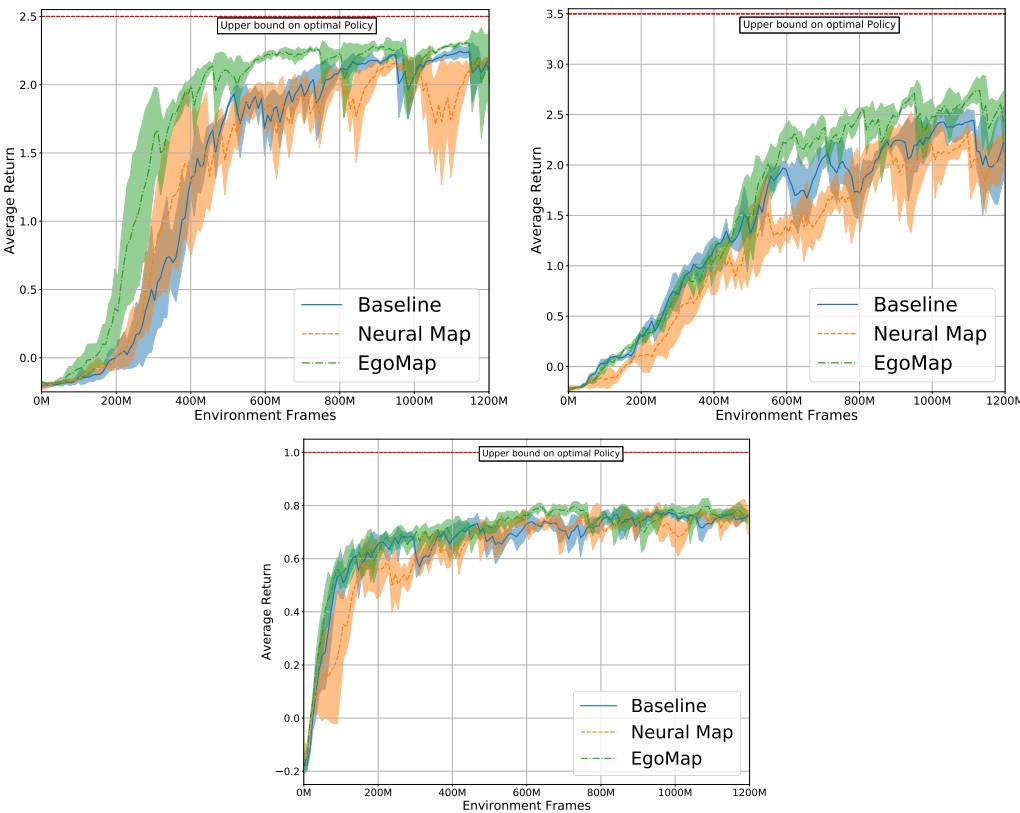

Figure 8: Training curves of held out test set performance on 4-item (top left), 6-item(top-right) and labyrinth scenarios (bottom).

