# OpenReview forum: "EgoMap: Projective mapping and structured egocentric memory for Deep RL"
_ICLR.cc/2020/Conference — Reject_

### Official Review · AnonReviewer3 · 2019-10-23
**Official Blind Review #3**

**Rating:** 6

**Review:**

The paper proposes a novel architecture for spatially structured memory. The main idea is to incorporate inductive bias/invariance derived from projective geometry arguments. The experiments seem to clearly show that this new architecture improves previous approaches to tasks which require spatial reasoning and memory, and the ablations studies and visualizations provide useful insights into the workings of the agent. One thing I'm missing is an experiment showing that this inductive bias also doesn't degrade performance on tasks where spatial reasoning is not necessary (as compared to vanilla GRU/LSTM).

**Experience Assessment:**

I do not know much about this area.

**Review Assessment: Checking Correctness Of Derivations And Theory:**

N/A

**Review Assessment: Checking Correctness Of Experiments:**

I assessed the sensibility of the experiments.

**Review Assessment: Thoroughness In Paper Reading:**

I made a quick assessment of this paper.

---

> ### Author Response · Authors · 2019-11-08
> **Response: Official Blind Review #3**
>
> Thank you for your review. Regarding your question about comparison on a task in a 3D environment that requires memory but not spatial reasoning, we found it challenging to create such a task as most problems in 3D environments require some sort of spatial reasoning. The closest in the suite of tasks we evaluated on is that of Labyrinth, where the agent must find an exit of a maze. This task does not require the agent to backtrack to a previously observed object / location and we observe comparable performance on this task between the baseline and EgoMap agent. However, we still see a small improvement in performance over the baseline agent, likely because the agent can store the wall locations to better localize itself and identify yet explored areas of the environment.

---

### Official Review · AnonReviewer2 · 2019-10-24
**Official Blind Review #2**

**Rating:** 3

**Review:**

This paper studies how to build semantic spatial maps for the purpose of navigation in 3D environments. The paper presents a differentiable policy network that pastes together semantic map predictions into a spatial map. Information is read out from this map using a global read operation (that looks at the entire map) and a self-attention read operation. This information is used to produce actions. The paper presents experimental results in 3D VizDoom scenarios and reports improvements over a vanilla LSTM, and another spatial memory based method (Neural Map).

Strengths:
1. I very much like the proposed formulation for tackling navigation problems. Using learning to leverage semantic reasoning, and structuring the computation spatially makes a lot of sense.
2. In my view, the proposed formulation advances current models in the following ways:
a. Maintaining and updating allocentric maps, and reading off egocentric maps. This alleviates need for repeated rotations of the map, and thus prevents aliasing.
3. The paper provides ablations for the various parts of the system and provides qualitative analysis of the learned spatial representations.
4. Very good placement of work in current literature. I really like Table 1.

Shortcomings:
1. The central contribution of the paper is the design of the egocentric spatial memory, how to build and maintain it over time, and its use in deep RL. The paper does this by using components from previous papers and presents a very nice summary of this in Table 1. Unfortunately, modulo the component described above (that of maintaining allocentric maps and reading off egocentric maps as and when needed), all other components are borrowed from existing papers, as can be seen in Table 1 already. The paper lists its contributions in Introduction on page 2, and each of those contributions has been studied in previous papers (though I will that admit no single paper does all these things together). Thus, I believe the paper falls short in terms of technical contributions.

2. Following on from point above, putting everything together and showing that it works, could also be a reasonable contribution, though it would warrant more extensive and systematic experiments for the different design choices, possibly in more realistic environments. For example, a) is the projective projection important, or could that have been learned, b) do repeated rotations indeed lead to blurred representations, c) what is critical to get such models work with RL, that past models that used imitation learning couldn't, d) other claimed differences from past works in this space.

3. Experiments and analysis:
a. The paper compares against NeuralMap, and reports improvements, but doesn't give a reason as to why this happens.
b. Past works have demonstrated these ideas in visually realistic environments (similar to those in Gibson / Habitat, see semantic tasks in CogMap). Current paper only investigates proposed ideas in VizDoom environments.

Thus, while I like the direction of research and the fact that the paper presents an architectures that uses latest techniques in the area, I believe the paper doesn't have enough technical contribution of its own, and experiments are limited to synthetic VizDoom environments.

**Experience Assessment:**

I have published in this field for several years.

**Review Assessment: Checking Correctness Of Derivations And Theory:**

N/A

**Review Assessment: Checking Correctness Of Experiments:**

I carefully checked the experiments.

**Review Assessment: Thoroughness In Paper Reading:**

I read the paper at least twice and used my best judgement in assessing the paper.

---

> ### Author Response · Authors · 2019-11-07
> **Response:  Official Blind Review #2**
>
> Thank you for providing such a detailed and constructive review. We have uploaded an updated version of the paper and will endeavour to address the shortcomings you have highlighted.
>
> 1. While some aspects of this work have been implemented in other works, we wanted to address their limitations such as discrete state spaces, imitation learning, simple tasks and supervised learning. As you have highlighted, some of the solutions we found in order to combine these approaches are novel. Such as allocentric map storage and egocentric reading, and perhaps should be highlighted in more detail.
>
> 2.
> a) Learning the projective transform: We think that this is possible, but not with a weak RL reward signal. The projection module in Cog. Map seems to learn an approximation of this directly, but requires a deep Resnet 50 encoder-decoder network and imitation learning. We are not aware of any works that train such deep networks from RL alone, the deepest being IMPALA [1].
>
> b) Blurred representations: We have updated the appendix (page 13) with synthetic examples of the blurring observed when we do not use an allocentric frame of reference for map storage.
>
> c) What is required to get this to work: RL is known to require small networks (in comparison to supervised learning) the inclusion of the inductive bias of projective geometry means we can bypass a large part of the learning process, which in other cases such as Cog. map, required deep networks and imitation learning.
> d) Other differences: See 3.a
>
> 3.
> a) Improvements over Neural Map: We added to the conclusions the following statement “The increase in performance compared to Neural Map is due to two aspects. 1) The differential projective transform maps what the objects are to where they are in the map, which allows for direct localization with attention queries and global reads. In comparison, Neural Map writes what the agent observes to where the agent is on the map, this means that the same object viewed from two different directions will be written to two different locations on the map, which leads to poorer localization of the object. 2) Neural Map splits the map to four 90-degree angles, which alleviates the blurring highlighted in the appendix, our novel solution to this issue stores a single unified map in an allocentric frame of reference and performs an offline egocentric read, which allows an agent to act in states spaces where the angle is continuous, without the need to quantize the agent’s angle to 90-degree increments.”
>
> b) Realistic environments: We briefly discussed the pros/cons of realistic simulators at the start of section 4. To elaborate, while there are a number of impressive looking simulators available (Gibson, Home, Habitat), the tasks available in these simulators are simple, or attempt to combine domains such as language and navigation. In this work we are focused on memory. To demonstrate the benefits of spatially structured memory we evaluated on tasks that require an agent to explore and revisit parts of the environment. Out of the box, the visually realistic simulators do not provide such challenging multi-step tasks and often to not provide the flexibility to design such tasks. Whereas more mature simulators such as ViZDoom and DeepMind lab provide tools for the design of novel tasks, or the tasks have already been created by the community.
>
>
> Refs.
> [1] Espeholt et al. IMPALA: Scalable Distributed Deep-RL with Importance Weighted Actor-Learner Architectures

---

### Official Review · AnonReviewer4 · 2019-11-02
**Official Blind Review #4**

**Rating:** 6

**Review:**

This very well written and executed paper synthesizes several ideas recently published in the field of deep reinforcement learning-based goal-driven navigation. It elegantly combines these ideas together by presenting a neural agent architecture that consists of:
* a perception module (e.g. a convnet) that extracts coarse visual feature maps s_t from an RGBD image
* a differentiable map canvas M_t that is rotated at each step based on affine egocentric velocity (dx_t, dy_y, d \phi_t)
* differentiable inverse projection mapping, which uses known camera parameters, projective geometry and the depth channel of the image to project the visual feature vectors s_t onto a 2D map and add it to the existing canvas M_t
* a recurrent module (GRU) for update a state h_t that is used for computing the policy distribution and value function
* additional inputs to the policy and value function, that include a global map read r_t, as well as a query q_t (produced by the policy head) based retrieval of features from the map
* position indexing of features retrieved from the map

The algorithm is trained end-to-end, without extra supervision, using Advantage Actor-Critic (A2C) RL. Based on the strong inductive biases regarding the map, namely affine transforms of the map given information about relative movement, and projective geometry transformations of visual features in the map frame, it seems that the question of where to write is solved, and that the network only needs to learn what to write in the differentiable map. Evaluation is done on 3 games in VizDoom: finding the exit of the Labyrinth, object retrieval and find and return / Minotaur.

Criticism:

The authors could justify better the choice of using the projective geometry inductive prior. They use sentences like "We argue that projective geometry is a strong law imposed on any vision system working from egocentric observations" (not quite related to grid and place cells, despite being in that section) and "this inverse mapping operation is second nature to many organisms" without giving any reference.

Several papers have been published in the last two years, focusing on differential memory architectures with a 2D map structure, projective geometry. This paper goes further by building and iteratively updating a 2D occupancy map using visual features and image geometry, just like RGBD-SLAM (which would merit a citation, e.g., [1] and [2]). This paper essentially combines existing ideas (see table 1): projective geometry, reward-based learning of M_t, RL, multitask navigation, semantic features. While this is not novel, seeing all this combined in a single technique does have merit.

What is disappointing, given that this is a combination paper, is that the environment is so simply, and that photorealistic environments were not tested. For example, the VizDoom environment uses 2D sprites for objects, making the visual feature extraction from objects much simpler. Would the method work equally well with the objects in DeepMind Lab, which are seen from multiple view points? And what in an environment like AdobeIndoorNav?

[1] Henry et al (2010) "RGB-D mapping: Using depth cameras for dense 3D modeling of indoor environments"
[2] Izadi et al (2011) "KinectFusion: real-time 3D reconstruction and interaction using a moving depth camera"

**Experience Assessment:**

I have published in this field for several years.

**Review Assessment: Checking Correctness Of Derivations And Theory:**

I carefully checked the derivations and theory.

**Review Assessment: Checking Correctness Of Experiments:**

I carefully checked the experiments.

**Review Assessment: Thoroughness In Paper Reading:**

I read the paper thoroughly.

---

> ### Author Response · Authors · 2019-11-08
> **Response: Official Blind Review #4**
>
> Thank you for taking the time to provide such detailed feedback, we have updated the paper based on your comments.
>
> Projective geometry prior: We have updated the related work section of the paper, and moved the sentence "We argue that projective geometry is a strong law imposed on any vision system working from egocentric observations", from the Grid cells section to the spatial memory section, which is now titled “Projective geometry and spatial memory”.
>
> We have also updated section 3, paragraph 2, which discusses the motivation behind the inverse mapping operation, to include a study on representations of 3D spaces of different species [1]. This includes analysis of depth prediction in mammals, the first step in the inverse projective mapping process. We would also like to point out that our Ego map is accompanied by a classical (flat vectorial) hidden state, which can store information without spatial meaning, for instance the fact whether a certain item has been seen or not, or whether it has been picked up or not.
>
> References on RGBD slam and KinectFusion, have been added to the related work in the section on projective geometry and spatial memory.
>
> Realistic environments: We briefly discussed the pros/cons of realistic simulators at the start of section 4. To elaborate, while there are a number of impressive looking simulators available (Gibson, Home, Habitat), the tasks available in these simulators are simple, or attempt to combine domains such as language and navigation. In this work we are focused on memory, in particular on learning affordances (objects, their positions and their relationship to the task) from reward. To demonstrate the benefits of spatially structured memory we focused on tasks that require an agent to explore and revisit parts of the environment. Out of the box, the visually realistic simulators do not provide such challenging multi-step tasks and often do not provide the flexibility to design such tasks. Whereas more mature simulators such as ViZDoom and DeepMind lab provide tools for the design of novel tasks, or the tasks have already been created by the community. You raise a good point regarding the 2D sprites in ViZDoom, we think that the recognition of 3D objects would be slightly more difficult, but neural networks certainly have the capacity to learn suitable internal representations. We also think the small increase in difficulty would have an equal effect on the baseline agent’s performance.
>
> The AdobeIndoorScenes simulator has a discrete state space and includes the Stanford large-scale 3D Indoor Spaces dataset, used in Cognitive Mapping and Planning. The downside of these grid based environments is that the agent’s turn actions are in 90 degree increments, which trivializes the affine transform of the spatial memory. In this work, we focused on an environment with a continuous state space.
>
> [1] Brain encoding and representation of 3D-space using different senses, in different species. Fregnac et al. 2004.

---

> > ### Comment · AnonReviewer4 · 2019-11-15
> > **Official Blind Review #4 update**
> >
> > Thank you for your response and for the additional references, in particular to projective geometry. As another reviewer pointed out, I still remain unconvinced by the simplistic environment which does not fully exploit the advantages of projective geometry, and believe that small-scale highly detailed environments and DeepMind lab would be more appropriate than VizDoom.

---

> > > ### Author Response · Authors · 2019-11-15
> > > **Response: Official Blind Review #4 update**
> > >
> > > Thank you for your further response. We understand the desire to push for more realistic environments. We believe that barring the 2D sprites you mentioned, the visuals of DeepMind Lab and ViZDoom are comparable. Both are 3D environments where an agent observes from a monocular viewpoint, and observations / features can be inverse projected in both simulators as they both provide a depth buffer and intrinsic camera properties. It is worth noting that works in this domain typically use RGB(D) inputs of height/width (84,84) [1], (64,96) [ours], (72,96) [2], (120,160) [3], sizes that are too small to capture the high fidelity details that may be available in photo-realistic environments. We think that on comparable tasks in DeepMind Lab our memory architecture should achieve a similar uplift in performance when compared with a standard LSTM based agent, in particular as the difference in performance is due to its capacity of representing spatial reasoning with a comparatively low number of parameters.
> > >
> > > We thank you again for your constructive feedback and our future works will be evaluated in more realistic (looking) environments.
> > >
> > >
> > >
> > > Refs:
> > > [1] Mirowski et al. Learning to Navigate in Cities Without a Map
> > > [2] Espeholt et al. IMPALA: Scalable Distributed Deep-RL with Importance Weighted Actor-Learner Architectures
> > > [3] Savinov et al. Episodic Curiosity Through Reachability

---

### Decision · Program_Chairs · 2019-12-19

**Decision:**

Reject

**Comment:**

This paper presents a spatially structured neural memory architecture that supports navigation tasks.  The paper describes a complex neural architecture that integrates visual information, camera parameters, egocentric velocities, and a differentiable 2D map canvas.  This structure is trained end-to-end with A2C in the VizDoom environment.  The strong inductive priors captured by these geometric transformations is demonstrated to be effective on navigation-related tasks in the experiments in this environment.

The reviewers found many strengths and a few weaknesses in this paper.  One strength is that the paper pulls together many related ideas in the mapping literature and combines them in one integrated system.  The reviewers liked the method's ability to leverage semantic reasoning and spatial computation.  They liked the careful updating of the maps and the use of projective geometry.

The reviewers were less convinced of the generality of this method.  The lack of realism in these simulated environments left the reviewers unconvinced that the benefits observed from using projective geometry in this setting will continue to hold in more realistic environments.   The use of fixed geometric transformations with RGBD inputs instead of learned transformations also makes this approach less general than a system that could handle RGB inputs.  Finally, the reviewers noted that the contributions of this paper are not well aligned with the paper's claims.

This paper is not yet ready for publication as the paper's claims and experiments were not sufficiently convincing to the reviewers.